# The Role of Anthocyanins in Plant Tolerance to Drought and Salt Stresses

**DOI:** 10.3390/plants12132558

**Published:** 2023-07-05

**Authors:** Siarhei A. Dabravolski, Stanislav V. Isayenkov

**Affiliations:** 1Department of Biotechnology Engineering, Braude Academic College of Engineering, Snunit 51, Karmiel 2161002, Israel; sergedobrowolski@gmail.com; 2Department of Plant Food Products and Biofortification, Institute of Food Biotechnology and Genomics, The National Academy of Sciences of Ukraine, Baidi-Vyshneveckogo Str., 2a, 04123 Kyiv, Ukraine

**Keywords:** anthocyanin, drought, salinity stress, abiotic stress, abscisic acid, microRNA, transcription factors (TFs)

## Abstract

Drought and salinity affect various biochemical and physiological processes in plants, inhibit plant growth, and significantly reduce productivity. The anthocyanin biosynthesis system represents one of the plant stress-tolerance mechanisms, activated by surplus reactive oxygen species. Anthocyanins act as ROS scavengers, protecting plants from oxidative damage and enhancing their sustainability. In this review, we focus on molecular and biochemical mechanisms underlying the role of anthocyanins in acquired tolerance to drought and salt stresses. Also, we discuss the role of abscisic acid and the abscisic-acid-miRNA156 regulatory node in the regulation of drought-induced anthocyanin production. Additionally, we summarise the available knowledge on transcription factors involved in anthocyanin biosynthesis and development of salt and drought tolerance. Finally, we discuss recent progress in the application of modern gene manipulation technologies in the development of anthocyanin-enriched plants with enhanced tolerance to drought and salt stresses.

## 1. Introduction

Anthocyanins are water-soluble blue, purple, or red pigments, which are mainly found in fruits, tubers, and flowers and are synthesised by the flavonoid metabolic pathway [1]. Usually, anthocyanins attract pollinators and facilitate seed spreading, protect from strong light and absorb ultraviolet light, scavenge ROS, and maintain osmotic balance. Structurally, anthocyanins are anthocyanidins in the form of glycosylation or acylation, with the six most abundant representatives in plants: delphinidin, peonidin, malvidin, cyanidin, pelargonidin, and petunidin [2].

Anthocyanins are synthesised in the cytosol and then modified to various anthocyanin derivatives and transported to vacuoles. In general, the main players in anthocyanin biosynthesis by the phenylpropanoid pathway are well studied. Briefly, the pathway starts with the amino acid phenylalanine, which is further modified and condensed with one molecule of 4-coumaroyl-coenzymeA and three molecules of malonyl-CoA, leading to the formation of naringenin chalcone. In the next step, the pathway diverges into sidebranches leading to different classes of flavonoids, including anthocyanins. The core genes involved in flavonoid biosynthesis are well studied and classified as early biosynthesis genes (EBGs) (chalcone synthase and isomerase (*CHS* and *CHI*), flavanone 3-hydroxylase and synthase (*F3H* and *FLS*), and others) and late biosynthesis genes (LBGs) (such as dihydroflavonol-4-reductase (*DFR*), uridine diphosphate-glucose: flavonoid 3-O-glucosyltransferas (*UFGT*), anthocyanin synthetase (*ANS*), and glutathione S-transferase (*GST*)) [3]. Competition between FLS and DFR regulates flavonol synthesis or anthocyanin accumulation, respectively. DFR is the key enzyme in anthocyanin biosynthesis, it converts dihydrokaempferol, dihydroquercetin, or dihydromyricetin (DHK, DHQ or DHM, respectively) into leucoanthocyanidins, which are further converted by ANS into other anthocyanidins, modified by glycosylation, acylation, and methylation (Figure 1) [4] and transported by GST and the multidrug and toxic compound extrusion (MATE) transport system into the vacuole [5,6,7]. There are hundreds of unique anthocyanins and many enzymes involved in their modifications; however, these late modification pathways are mostly unknown [8]. The transcriptional regulation of anthocyanin late biosynthesis genes and transporters is mostly conducted by three types of transcription factors (TFs)—myeloblastosis (MYB), basic helix-loop-helix (bHLH), and WD40, which subsequently directly activate/repress other structural genes and downstream TFs [9,10].

Drought is one of the major abiotic stresses and is considered a global threat to world food security through negative effects on plant growth and productivity [11]. Drought is mainly associated with osmotic stress leading to the disruption of cell division and proliferation, interruption of normal growth and development, and increasing ROS generation, which in total, results in reduced quality and quantity of yield [12]. In recent decades, several research papers have suggested that drought-mediated induction of anthocyanins may be used for osmoregulation and scavenging surplus ROS [13,14]. Salinity represents another serious abiotic stressor, which combines osmotic and ionic stresses. Disrupted Na^+^ and Cl^−^ homeostasis leads to the limitation of water and mineral uptake, damages cellular membranes, and decreases the efficiency of photosynthesis and crop yield [15]. Similarly to their role in drought stress response, salt-stress-induced anthocyanins have been suggested to reduce ionic and osmotic damage and scavenge ROS [16,17]. However, only the application of recently invented omics technologies and gene editing tools allowed us to investigate the exact molecular mechanisms underlying anthocyanin-mediated beneficial effects on plant drought and salt tolerance/resistance.

In this review, we focus on the most recent works dedicated to various mechanisms associated with anthocyanin biosynthesis and tolerance to drought and salt stress. The regulation of stress-induced anthocyanin biosynthesis by different TFs is discussed. The physiological and biochemical responses with the involvement of anthocyanin biosynthesis are also briefly summarised. Therefore, this review may provide new insights for future researchers studying the role of the anthocyanin biosynthesis pathway in stress tolerance. Additionally, we discussed recently discovered regulatory mechanisms connecting stress-induced anthocyanin accumulation to hormonal signalling pathways, which can further accelerate research on the development of new stress-tolerant, biofortified varieties and breeding new cultivars.

## 2. Anthocyanins and Drought Tolerance

### 2.1. Anthocyanins Induction by Drought

Anthocyanins are naturally occurring compounds which are synthesised in response to various environmental stresses (both abiotic and biotic, such as low temperatures, salt and drought, nutritional deficiency, pathogen attacks, and heavy metals) and grant various colours (such as purple, red, and blue) to different plant tissues and organs [18]. For example, as was shown on the mature calyxes of several cultivars of *Hibiscus sabdariffa* (L.), the total anthocyanin content reached the highest concentration at a 65% moisture irrigation regime. In particular, the content of cyanidin 3-O-sambubioside (by 55% relative to the control level) cyanidin, cyanidin 3-O-sambubioside, cyanidin 3-O-glucoside, and delphinidin 3-O-glucoside were greatly increased. While the contents of these compounds are well correlated with the colour of the calyxes, the severe water stress (under 33% moisture) resulted in decreased anthocyanin contents in all of the *Hibiscus* cultivars [19].

Recently, the functional involvement of anthocyanins in defence against the deleterious effects of drought was investigated on the purple-stem genotype of *Brassica napus* (L.) [20]. In general, the purple-stem genotype showed higher antioxidant capability and drought stress tolerance in comparison with the green-stem genotype. Accordingly, the expression and activity of antioxidant enzymes and the content of proline and soluble sugar were higher in the purple-stem genotype, which allow them to more effectively alleviate drought-stress-induced increased accumulation of ROS and MDA. Furthermore, the purple-stem genotype under drought showed higher chlorophyll production, photosynthesis rate, photosystem II quantum yield, electron transport rate (II) efficacy and higher photochemical protective mechanisms, and higher biomass and relative water content than the green-stem genotype, thus suggesting that the purple-stem genotype is more resistant to drought [20].

In addition, an investigation of transcriptomic and metabolomic responses of millet (*Panicum miliaceum* L.) using drought-resistant ‘Hequ Red millet’ and drought-sensitive ‘Yanshu No. 10’ varieties identified 97 structural genes associated with anthocyanin biosynthesis after 6 h of drought stress [21]. Interestingly, 25 *GST*, 9 *4CL*, 8 *F3H*, 4 *PAL*, 2 *CHS*, 2 *CHI*, 1 *DFR,* and 1 *OMT* genes were significantly up-regulated in both varieties, and 8 *GST*, 3 *F3H*, 2 *OMT,* and 1 *4CL* genes were up-regulated only in the ‘Hequ Red millet’ variety. These data indicate that while both varieties significantly enhanced the metabolic activity of anthocyanin biosynthesis when subjected to drought, the set of anthocyanin-metabolism-related genes activated specifically in the ‘Hequ Red millet’ variety play a crucial role in its drought-resistant phenotype [21].

Similar results were obtained on tea (*Camellia sinensis* (L.) Kuntze), where high soil water deficit was associated with the depression of LBGs and suppression of anthocyanin transport genes [22]. On the contrary, purple-pigmented leaves sampled during wet conditions revealed higher expression of 63 TFs regulating anthocyanin biosynthesis (including 8 *MYBs*, 8 *bHLHs*, 22 *WD40,* and others), biosynthesis genes themselves (such as *CHS1* and *2*, *F3H*, *F3′5′H*, *PAL4*, *DFR*, *ANR*, *CHI3,* and others), and anthocyanin transporters (including 39 genes from *ABC* family, 17 *GST,* and 4 *MATE*), which well correlated with higher anthocyanin accumulation. Accordingly, anthocyanins were used as co-substrates by antioxidant enzymes (specifically, by SOD and CAT) to scavenge surplus ROS released from organelles during the dry season [22].

Investigation of a purple variety, B100, of foxtail millet (*Setaria italica* (L.) P. Beauvois) revealed higher accumulation of anthocyanin in leaf epidermis, antioxidant capacity, and enhanced stress resistance in comparison with the green-leaved variety, YG1, while the contents of chlorophyll and carotenoids were higher in YG1. These results were supported by transcriptome analysis and qRT-PCR, which showed up-regulated expression of nine genes involved in anthocyanin biosynthesis (*PAL*, *4CL*, *DFR*, *LDOX1* and *2*, *AT*, *UFGT*, *GT,* and *5GT*), thus confirming the close association between anthocyanin biosynthesis/accumulation and stress resistance [23]. Interestingly, a comparison of purple, blue, black, and white wheat lines revealed the role of *F3H* and *F3′5′H* genes as the most significant influencers in the development of purple and blue colours, respectively [24].

In total, these results suggested that the production and accumulation of anthocyanins are increased under drought stress. Subsequently, anthocyanin accumulation resulted in improved content of osmoprotectors, increased ROS scavenging and antioxidant capacities, and preserved photosynthetic apparatus and cellular components from oxidative damage, thus facilitating higher tolerance to drought. However, further research is required to functionally characterise genes and molecular pathways involved in physiological and metabolic responses observed in coloured drought-tolerant/resistant species.

### 2.2. The Role of Abscisic Acid in the Regulation of Anthocyanin Biosynthesis

The plant hormone abscisic acid (ABA) is known to play an important role in the regulation of plant water balance by stomatal closure during drought. The expression of genes involved in ABA biosynthesis is up-regulated by drought, with further binding to the ABA receptors and increasing levels of ROS and Ca^2+^ in guard cells (Figure 2). Subsequently, ROS and calcium modulate the activities of ion channels, decreasing guard cell turgor, and closing the stomata [25].

Recently, the molecular mechanisms of ABA involvement in the regulation of anthocyanin biosynthesis have been suggested. As was shown on *Aristotelia chilensis* (Molina) Stuntz), severe drought significantly up-regulated both sets of genes responsible for ABA and anthocyanin biosynthesis. In particular, levels of 9-cis-epoxycarotenoid dioxygenase (*NCED*) and *UFGT* correlated with ABA and total anthocyanin levels, respectively [26]. Furthermore, the application of fluridone (an ABA biosynthesis inhibitor) on drought-stressed *A. chilensis* plants greatly decreased both ABA (about 75%) and total anthocyanin (about five-fold) concentrations. Accordingly, ABA treatment restored ABA and total anthocyanin levels in stressed plants. Also, the expression level of the *AcUFGT* gene decreased upon the fluridone treatment and increased after ABA application, thus confirming the involvement of ABA in the regulation of anthocyanin biosynthesis under drought [27]. Similar results were demonstrated on drought-stressed tea (*Camellia sinensis* (L.) Kuntze) plants, where exogenous ABA treatment up-regulated metabolic pathways related to the amino acid, lipid, and energy metabolism and flavonoid biosynthesis and, subsequently, resulted in increased contents of flavonols, flavones, isoflavones, and anthocyanins [28].

#### Abscisic Acid-microRNA Regulatory Module

The research evidence accumulated during the last decade suggests that ABA can regulate stress tolerance and anthocyanin biosynthesis through the regulation of microRNAs (miRNAs) [29]. Plant miRNAs are small non-coding RNAs (around 20–24 nucleotides) that regulate gene expression via transcriptional or post-transcriptional pathways. In particular, miRNA156 has been shown to interact with the squamosa promoter binding protein-like (*SPL*) gene and regulated various developmental and physiological responses to environmental stimuli [30]. Furthermore, the ABA-miRNA156 regulatory module was associated with an increased accumulation of anthocyanins in drought-resistant plants (Figure 2) [31].

Thus, expression of pre-miR156e-3p from *Paeonia lactiflora* Pall. in Arabidopsis resulted in down-regulation of the *SPL1* gene, which is a known negative regulator of anthocyanin biosynthesis. The expression level of the *DFR* gene, a downstream target for *SPL1*, was strongly up-regulated in transgenic plants, which resulted in improved anthocyanin accumulation and the appearance of purple colour on lateral branches [32]. Silencing of *SPL9* with RNAi in lucerne plants (*Medicago sativa* L.) affected the phenotype in normal watering conditions (reduced stem thickness and plant height). However, under drought conditions, transgenic plants showed a lower rate of leaf senescence and improved relative water content, thus suggesting greater drought stress tolerance. Furthermore, SPL9-RNAi transgenic plants have accumulated more anthocyanin in both conditions (well watered and drought stressed), suggesting that *SPL9* is involved in the regulation of anthocyanin biosynthesis [33].

Similar results have been shown in lucerne plants overexpressing *miRNA156*, where improved drought tolerance was associated with increased ABA and antioxidant levels, and enhanced proline accumulation. At the same time, miR156OE plants showed reduced expression of the miR156-targeted gene *SPL13* and demonstrated reduced water loss and enhanced chlorophyll content, photosynthetic assimilation, and stomatal conductance [34]. Further, the involvement of the WD40-1 (a positive regulator of DFR gene expression) in the miR156-SPL13 interplay was shown (Figure 2). In the miR156OE lines, the expression of *SPL13* was suppressed, while the expression of *WD40-1* and *DFR* was increased and linked to enhanced anthocyanin biosynthesis. Interestingly, SPL13 was demonstrated to directly bind the promoter region of the *DFR* gene in vivo to regulate its expression [35].

Recently, experiments on mutant cassava (*Manihot esculenta* Crantz) plants overexpressing a dominant-negative form of miR156-resistant *MeSPL9* demonstrated increased accumulation of osmoprotective metabolites (proline and soluble sugars), anthocyanin, and endogenous jasmonic acid. These results have been confirmed with transcriptomic analysis, which identified differentially expressed genes related to sugar and JA signalling and metabolism in transgenic cassava plants under drought stress conditions [36]. Interestingly, other players in the miR156-SPL-anthocyanin axis were identified in transgenic poplar (*Populus alba*×*P. tremula* var. glandulosa) expressing *miR156* from *Medicago truncatula*. In particular, levels of *miR160h* and *miR858* positively correlated with *miR156* levels, suggesting their interaction with the miR156-SPL module and potential role in anthocyanin accumulation through regulation of auxin response factors and MYB TFs, respectively. Similar to results from other species, transgenic poplar plants showed increased levels of flavones, flavonols, and anthocyanins, while the level of total lignin content was reduced [37].

Taken together, these results indicate that ABA is a crucial regulator of anthocyanin biosynthesis under drought stress. Furthermore, ABA interacted with many other factors such as microRNAs, TFs, and structural genes, which create a complex regulatory network to fine-tune the process of drought-mediated anthocyanin biosynthesis. While several other hormonal pathways have been shown to be briefly affected (JA and auxin), their role in the regulation of drought-mediated anthocyanin biosynthesis is currently unknown and requires further investigation.

### 2.3. Transcription Factors Regulating Drought-Mediated Anthocyanin Biosynthesis

Anthocyanin biosynthesis (in both normal- and drought-stressed conditions) is regulated through sophisticated transcriptional regulatory networks. In addition to the previously mentioned complexes of the MYB-bHLH-WDR (MBW) TFs regulating anthocyanin late biosynthesis genes and transporters [10], many other different types of TFs have been found to fine-tune anthocyanin homeostasis (Figure 3). Further in this section, we focus on the recent advances in the characterisation of the various TFs involved in the regulation of drought-mediated anthocyanin biosynthesis and accumulation.

#### 2.3.1. Myeloblastosis Transcription Factors

The MYB TFs are one of the major TF families in the plant kingdom in general and in the regulation of anthocyanin homeostasis. Their significant role has been demonstrated in many plant species [38]. For example, recent RNA-seq analysis of the MYB TFs between the pigmented (white, red, and purple) and between the drought-tolerant and drought-sensitive cultivars of potato (*Solanum tuberosum* L.) have identified 19 StMYB-related TFs that were differentially expressed between pigmented and white skin and flesh, and 26 StMYB-related proteins that were differentially expressed between the drought-tolerant and drought-sensitive cultivars under drought. Such complex networks of differentially expressed StMYB-related proteins in various potato cultivars confirmed their significant functional involvement in drought-mediated anthocyanin biosynthesis [39].

Interestingly, among 272 MYB family TFs identified in *Carthamus tinctorius* L., the expression of *CtMYB63* was increased under drought and cold stress (Figure 3). Furthermore, expression of *CtMYB63* in *A. thaliana* resulted in enhanced root growth, up-regulation of anthocyanin biosynthesis genes (*PAL*, *4CL*, *CHS*, *CHI*, *F3H*, *F3′H*, *DFR*, and *ANS*) and, subsequently, higher anthocyanin accumulation. Therefore, these data suggest that *CtMYB63* is involved in abiotic stress tolerance through increased anthocyanin biosynthesis and accumulation [40]. Similarly, the *MYB24* gene, isolated from drought-tolerant Chinese wild *Vitis yanshanesis*, was shown to play an important regulatory role in the biosynthesis of proanthocyanidin, anthocyanin, and flavonoid. The expression of *VyMYB24* in tobacco resulted in dwarf phenotype, reduced content of gibberellin (GA1+3), and improved tolerance to drought. Furthermore, mutant plants accumulated more proline and demonstrated enhanced expression of antioxidant genes (*SOD*, *POD*, and *CAT*) and increased activity of these enzymes. Interestingly, spraying with exogenous GA partly recovered the dwarf phenotype of mutant plants, suggesting the involvement of *VyMYB24* in the regulation of GA metabolism. Their results indicate that *VyMYB24* is a positive regulator of drought stress response and a crucial player in plant development via its involvement in GA metabolism [41].

Recent research has reported the MYB6-based mechanism of drought-induced anthocyanin accumulation in petals of *Chaenomeles speciosa* (Sweet) Nak.), which is known to change colour under drought. Experiments with transient expression and suppression of *CsMYB6* demonstrated that it is the key TF regulating anthocyanin biosynthesis genes (*ANS*, *DFR*, and *UFGT*) and co-regulatory TF (*bHLH111*) = that increased the promoter-binding ability of *CsMYB6* to anthocyanin structural genes [42] (Figure 3).

#### 2.3.2. Basic Helix-Loop-Helix Transcription Factors

The *Lc* gene from maize (*Zea mays* L.) is a well-studied bHLH TF involved in tissue-specific anthocyanin production [43]. Recently, the expression of the *Lc* gene in bamboo (*Dendrocalamus latiforus* Munro) was used to generate non-green bamboo with enhanced tolerance to abiotic stresses. The expression levels of nine genes involved in anthocyanin biosynthesis (*CHS*, *CHI*, *F3H*, *F3′5′H*, *F3′H*, *DFR*, *ANS*, *UFGT*, and *3GT*) were up-regulated in transgenic plants, which resulted in enhanced accumulation of anthocyanins (peonidin 3-O-rutinoside and cyanidin-3-O-rutinoside) and purple colouring (Figure 3). Also, transgenic plants showed higher tolerance to drought and cold stress, and increased levels of jasmonic acid, while levels of ABA and SA were not affected [44]. Additionally, SPATULA (SPT), another bHLH family TF which is known to regulate various processes in Arabidopsis (such as trichome and stomata formation, control of leaf size, floral transition and others) [45,46], was recently also demonstrated to regulate sucrose-mediated anthocyanin biosynthesis [47].

#### 2.3.3. Apetala2/Ethylene Responsive Factor

Expression profiles of *AP2/ERF* (APETALA2/ETHYLENE RESPONSIVE FACTOR) genes in eggplant (*Solanum melongena* L.) suggested that they are involved in responses to various abiotic stresses (cold, drought, and salt), hormone signalling pathways (ABA and ethylene), and in the regulation of anthocyanin biosynthesis [48]. The underlying molecular mechanism was characterised in apple (*Malus* × *domestic*), where MdERF38 interacted with MdMYB1, a known regulator of anthocyanin biosynthesis and drought tolerance [49]. Accordingly, the expression of *MdERF38* was enhanced under drought, while its proteasomal degradation was reduced. Furthermore, the transgenic calli overexpressing *MdERF38* accumulated more anthocyanin, and the expression of anthocyanin biosynthesis genes was up-regulated (*MdCHS*, *MdCHI*, *MdUF3GT*, and *MdDFR*). Interestingly, the binding strength of MdMYB1 to the promoters of *MdDFR* and *MdUF3GT* was enhanced with the addition of MdERF38, suggesting that MdERF38 facilitates the transcriptional activity of *MdMYB1*, thus promoting anthocyanin biosynthesis in response to drought [50].

#### 2.3.4. Other Transcription Factor Families

WRKY. The involvement of WRKY family TFs in drought and salt tolerance and anthocyanin biosynthesis was demonstrated on a well-known Chinese traditional medicinal plant *Lycoris radiata* (L’Hér.) Herb [51]. From 31 of the total identified WRKY genes, the expression of *LrWRKY5*, *LrWRKY22*, and *LrWRKY23*, and *LrWRKY4* and *LrWRKY8* was significantly up-regulated by drought in roots and leaves, respectively. Interestingly, eight TFs defined as positive regulators of the anthocyanin biosynthesis (*LrWRKY1*, *LrWRKY3*, *LrWRKY6*, *LrWRKY7*, *LrWRKY13*, *LrWRKY15*, *LrWRKY17*, and *LrWRKY27*), whereas four genes (*LrWRKY16*, *LrWRKY22*, *LrWRKY25*, and *LrWRKY29*) acted as negative regulators of the anthocyanin metabolites. These data suggest that *LrWRKY22*, which participated in both drought stress response and regulation of anthocyanin biosynthesis, is a potential target for future exploration [51] (Figure 3).

CBF. The expression of Arabidopsis TF from the dehydration-responsive element binding factors 1 (DREB1A)/C-repeat-binding factor (CBF3) family in cassava (*Manihot esculenta* Crantz) resulted in increased tolerance to cold and drought stresses [52]. However, mutant plants showed retarded growth, leaf curling, reduced storage root yield, and reduced anthocyanin accumulation. The expression of TFs associated with anthocyanin production (*EGL1*/*GL3*, *PAP2*, *PAP1*, *MYB4*, and *TTG1*) and anthocyanin biosynthesis genes (*PAL*, *4CL*, *C4H*, *CHS*, *CHI*, *F3H*, *F3′H*, *F3′5′H*, *DFR,* and *ANS*) was down-regulated in mutant plans (with the only exception for *BZ1*, which was up-regulated) and resulted in reduced anthocyanin accumulation (Figure 3). Thus, *CBF3* improved cassava tolerance to droughts and cold stresses, while acting as a negative regulator of the anthocyanin metabolism [52].

NAC. Similarly, expression of *NAC019* TF from cabbage (*Brassica oleracea* L.) in *A. thaliana* resulted in a decreased tolerance to drought, reduced survival rate, lower ABA and proline content, and higher water loss rate [53]. Also, under drought, the content of antioxidant enzymes and anthocyanin was decreased in mutant plants, while the level of accumulated ROS was markedly increased. Furthermore, the expressions of anthocyanin biosynthesis genes (*PAL*, *C4H*, *CHS*, *F3H*, *ANS*, and *UFGT*) and corresponding anthocyanin biosynthesis regulating TFs (*TT2*, *MYB113*, and *TT8*) and ABA signalling genes were down-regulated in mutant plants under drought stress conditions, whereas the expression of ABA catabolism genes was up-regulated in mutant plants in both normal and drought conditions [53] (Figure 3).

СССH. Recently, the role of the CCCH type TF C3H35 in drought stress response was demonstrated in the roots of poplar (*Populus ussuriensis* Kom.). *PuC3H35* targeted *PuANR* and *PuEARLI1*, which are involved in the regulation of drought-stress-mediated biosynthesis of proanthocyanidin and lignin in the *P. ussuriensis* root (Figure 3). Overexpression of *PuC3H35* promoted biosynthesis of proanthocyanidin and lignin, which improved anti-oxidation properties, provided mechanical support, and facilitated vascular tissue development, resulting in enhanced tolerance to drought [54].

### 2.4. Other Types of Regulators

Besides TFs, many other genes have been shown to regulate anthocyanin metabolism and drought tolerance. For example, the Calmodulin-binding protein (CBP60g), which is known to participate in pathogen resistance and drought tolerance [55], was also shown to act as a negative regulator of anthocyanin biosynthesis. In particular, CBP60g down-regulated anthocyanin biosynthesis (*CHS*, *CHI*, and *DFR*) and regulatory TFs (*PAP1* and *TT8*) genes under drought stress treatment. However, the exact regulatory mechanism of the CBP60g on anthocyanin biosynthesis and drought tolerance is still poorly understood and requires further investigation [56]. Similarly, the expression of flavonol synthase (*FLS1*) from *Dendrobium officinale* (Kimura and Migo) in *A. thaliana* resulted in increased flavonol content, while the expression of *DFR* and *ANS*, and subsequently, the anthocyanin content, were decreased. Additionally, the expression of *DoFLS1* was up-regulated after drought, cold, and salicylic acid treatments, suggesting the involvement of *DoFLS1* in the anthocyanin biosynthesis response to abiotic stresses [57].

The Psb28, a soluble protein in the photosystem II (PSII) complex, acted as a positive regulator of drought tolerance in wheat (*Triticum aestivum* L.) through increased expression of the chlorophyll synthase (*ChlG*) gene, and higher chlorophyll and lower MDA contents [58]. In wild-type plants, drought increased ABA and zeatin content, up-regulated expression of *RD22*, *DFR*, and *ANR*, and enhanced content of proanthocyanidins, delphinidin and cyanidin, delphinidin, and proanthocyanidins. However, in the expression of the wheat *Psb28* gene in transgenic *Arabidopsis* plants, the content of ABA and zeatin were restored to the control level under drought stress treatment and stomatal closure was promoted. These data suggest that *Psb28* exerted a positive function in the drought response by influencing the metabolism of ABA and zeatin hormones, which acted through stomatal closure and promoted anthocyanin accumulation [58].

Interesting results were shown on grape (*Vitis vinifera* L.) leaves infected with grapevine leafroll disease (GLD), common symptoms of which are downward rolling of leaf margins and enhanced biosynthesis of flavonoids and anthocyanins, which caused reddish-purple colouration on the leaf blades in [59]. Therefore, the effect of drought on wild-type leaves did not affect anthocyanin content, while leaves infected with GLD resulted in increased accumulation of total anthocyanin (11 peaks for different compounds) with or without drought. GLR infection up-regulated the expression of anthocyanin biosynthesis-related genes (*MYBA1* and *UFGT*), while drought further enhanced the expression of these genes, thus facilitating anthocyanin accumulation [60].

In summary, in recent decades many TFs from different families regulating drought-mediated anthocyanin production have been identified. While some TFs acted as positive or negative regulators (such as VyMYB24 and BoNAC019, respectively) of both anthocyanin biosynthesis and drought tolerance, others provide more specific negative effects on anthocyanin biosynthesis but increase drought stress tolerance (such as CBF3). Further research is required to better characterise cross-talk between the anthocyanin biosynthesis/drought stress tolerance axis with other signalling systems and physiological processes (such as ABA and JA hormonal signalling pathways, lignin metabolism, photosynthetic apparatus, and other types of stress).

## 3. Anthocyanins and Salinity Stress

### 3.1. Anthocyanin Induction by Salinity Stress

Salt stress disrupts plant physiology and metabolism through ion toxicity, oxidative and osmotic stress, and nutrient imbalance, leading to cellular damage and reducing growth and yield and, in severe conditions, even plant death [61]. Similarly to drought, salinity stress increases the capacity of the antioxidative system to neutralise the salt-stress-induced increase in surplus ROS production. For example, in *Hyssopus officinalis* L. plants the activities of antioxidant enzymes (CAT, SOD, and peroxidase) and contents of proline, phenol, and anthocyanin were significantly increased upon salt stress treatment [62]. Furthermore, several halophytic plant species are considered a valuable source of nutraceuticals such as 3-O-β-d-glucopyranoside flavonoid used for anti-obesity treatments [63,64,65].

Comparison of two accessions of the centipedegrass (*Eremochloa ophiuroides* (Munro) Hack.) demonstrated that salinity treatment inhibited leaf growth and decreased photosynthesis, chlorophyll content, and the maximal photochemical efficiency of PSII (Fv/Fm) in green-stem accession (E092-1), while the effect on the purple-stem accession (E092) was less pronounced. Also, the purple-stem accession had up-regulated levels of proline and antioxidant enzymes upon exposure to salinity conditions [66]. Similarly, a comparison of different coloured genotypes of wheat (*Triticum* sp.) showed higher anthocyanins and proline accumulation than in control plants after NaCl treatment. Also, coloured genotypes showed a better capacity to maintain the low accumulation of Na^+^, higher shoot K^+^ concentrations, and higher dry matter production after salt stress treatment [67].

Additionally, anthocyanin was shown to help plants cope with salt stress under nitrate-deficient conditions. Double mutant *Arabidopsis* plants *PAP1-D*/*fls1ko* with overexpressed *Production of Anthocyanin Pigment 1* (*PAP1*) gene and knockdown of *flavonol synthase 1* (*FLS1*) produce and accumulate high levels of anthocyanins because *PAP1* TF is a positive regulator of anthocyanin biosynthesis, and mutation in *FLS1* blocked production of the flavonol pathway [68]. Thus, *PAP1-D*/*fls1ko* plants efficiently absorbed and reduced nitrate, providing enhanced tolerance to nitrate-deficient salt stress conditions in comparison with wild-type plants or *ttg1* mutant plants, which have blocked anthocyanin synthesis. Also, genes related to nitrate metabolism (such as *NRT1.1*, *NiA1,* and *NiA2*) were up-regulated in *PAP1-D*/*fls1ko* plants [69].

At the same time, high NO_3_^−^ concentrations increase wild-type plants’ sensitivity to salt stress. However, both mutants (the *pap1-D*/*fls1ko* and *fls1ko*) accumulated higher levels of anthocyanin and showed higher ROS scavenging activities than wild-type plants under both normal and salt stress conditions. Apparently, enhanced nitrate reductase activities and expression of *NIA1* and *NIA2* in *pap1-D*/*fls1ko* and *fls1ko* plants increased the biosynthesis of proteins and proline, which strengthened salt stress tolerance [70]. Similarly, the relation between anthocyanin metabolism and nitrogen was demonstrated in carrot (*Daucus carota* L.) cell culture, which was subjected to NH_4_NO_3_/KNO_3_ salt stress in a bioreactor condition. The expression of anthocyanin biosynthesis (*UFGT*, *CHS*, *C4H*, and *LDOX*) and transporter (*MATE*) genes were up-regulated upon salt treatment, which resulted in increased anthocyanin production and its MATE-mediated transport to the vacuole. These data suggest that various salts (in addition to common NaCl) may induce anthocyanin production and vacuolar accumulation to alleviate the negative effects of salt stress [71].

On the other hand, opposite results have been reported for salt-tolerant rice genotypes with green (Pokkali) and purple (Niew Dam) leaves [72]. Both genotypes provided active physiological responses (osmolyte accumulation, antioxidant capacity, chlorophyll content, Na^+^/K^+^ ratio, osmotic adjustment, and membrane damage) and expression of salt stress-responsive ion transporters (*NHX1*, *SOS1-3*). However, the green-leaved genotype demonstrated lower Na^+^ accumulation in leaves when compared with the purple-leaved genotype, suggesting a better Na^+^ exclusion mechanism. On the other hand, the purple genotype accumulated a higher concentration of osmolytes, which resulted in enhanced osmotic adjustment. Interestingly, salt stress did not affect anthocyanin levels in the green genotype, while a reduction was identified in the purple genotype. These results suggested that green and purple genotypes utilise different salt tolerance mechanisms—Na^+^ ion exclusion and osmotic adjustment, respectively. Furthermore, these data demonstrate that anthocyanin played no vital role in salt stress tolerance in studied rice genotypes [72].

### 3.2. Transcription Factors Involved in Salt-Stress-Induced Anthocyanin Biosynthesis

#### 3.2.1. Myeloblastosis Transcription Factors

MYB TFs are important plant regulators of anthocyanin biosynthesis in normal and stress-mediated conditions [38]. Similarly to drought, this is one of the major families of TFs regulating the salt-stress-induced increase in anthocyanin production and accumulation. For example, *MYB90* from extreme halophyte species *Eutrema salsugineum* ((Pall.) Al-Shehbaz and Warwick), which has demonstrated a high degree of tolerance to multiple abiotic stresses, was shown as a major regulator of flavonoid biosynthesis (also including anthocyanins) [73]. Thus, the expression of *EsMYB90* in tobacco and *Arabidopsis* promoted the expression of early (*PAL*, *CHS*, and *CHI*) and late (*DFR*, *ANS*, and *UFGT*) anthocyanin biosynthesis genes and resulted in enhanced pigmentation and anthocyanin accumulation in stems, leaves, and flowers of transgenic plants [74]. Similarly, the expression of *EsMYB90* in wheat increased salt stress tolerance through enhanced POD and GST activity and reduced MDA content (Figure 4). Also, *EsMYB90* up-regulated a wide range of genes related to phenylpropanoid biosynthesis and antioxidant activity in transgenic plants subjected to NaCl treatment. Particularly, EsMYB90 bonded with the MYB-binding elements on the promoters of *TaANS2* and *TaDFR1* genes, thus activating their transcription and promoting anthocyanin biosynthesis in transgenic wheat plants [75].

Taxonomic affiliation: Ta (*Triticum aestivum* L.); Md (*Malus* × *domestic*); Es (*Eutrema salsugineum* (Pall.) Al-Shehbaz and Warwick)); and TFs with no affiliation belong to *Arabidopsis thaliana* (L.) Heynh.

The expression of the MYB family TF *PURPLE PLANT1/COLORED ALEURONE1* (*PL1*/*C1*) from wheat in *Arabidopsis* resulted in up-regulation of both early (*CHS*) and late (*DFR*, *LDOX*, and *UF3GT*) anthocyanin biosynthesis genes, which lead to higher anthocyanin accumulation in leaves (Figure 4). While in *pap1* mutants, the expression levels of structural genes and anthocyanin biosynthesis/accumulation did not change significantly. At the same time, environmental stresses (such as light, cold, and salt) greatly up-regulated *TaPL1* expression and induced anthocyanin accumulation. These results confirmed the role of TaPL1 as a main regulator of anthocyanin biosynthesis in wheat [76]. Several other MYB family TFs were identified as regulators of anthocyanin biosynthesis in *Arabidopsis* under salt and high light stress. The overexpression of *MYB112* caused up-regulation of biosynthesis genes (*DFR* and *ANS*) and TFs (*MYB114*, *MYB7*, *MYB32,* and *PAP1*) associated with anthocyanin production, while down-regulated TFs involved in flavonol biosynthesis (*MYB12* and *MYB111*) thus resulted in increased anthocyanin accumulation [77].

Recently, Arabidopsis *MYB3* TF was recognised as a repressor of salt-stress-mediated lignin and anthocyanin accumulation. Salt stress greatly up-regulated *MYB3* expression in roots, flowers, stems, and leaves. Therefore, *myb3* plants demonstrated enhanced expression of lignin and anthocyanin biosynthesis genes (*PAL1*, *C4H*, *COMT*, *4CL3*, *DFR*, and *LDOX*), higher lignin and anthocyanin accumulation and longer root growth under NaCl treatment when compared with wild-type plants. In particular, MYB3 interacted with TT8 and EGL3 TFs regulating anthocyanin biosynthesis [78]. Also, another study demonstrated that the negative regulation of jasmonate (JA)-responsive anthocyanin accumulation is mediated through the EAR motif-Containing Adaptor Protein (ECAP), JASMONATE-ZIM DOMAIN (JAZ6/8), and TOPLESS-RELATED 2 (TPR2) transcriptional repressor complex which represses the MBW complex [79]. As it was shown, moderate salt stress led to JA-dependent JAZ degradation, while high salinity stress induced the JA-independent degradation of ECAP through the 26S proteasome pathway. In the absence of salt stress conditions, ECAP directly binds and represses *MYB75*—one of the TFs from the MBW complex responsible for anthocyanin biosynthesis. Therefore, ECAP degradation causes *MYB75* activation and induces anthocyanin biosynthesis and accumulation. These results demonstrated that plants have several different strategies to regulate anthocyanin production under different severities of salt stress [80].

#### 3.2.2. Other Types of Transcription Factors

DRB. Double-stranded RNA-binding proteins (DRB1-DRB5) are important regulators of abiotic stress responses through the RNA silencing pathway and miRNA biogenesis [81]. Accordingly, *DRB2* and *DRB3* overexpressing *Arabidopsis* plants demonstrated salt-stress-tolerant phenotypes. On the contrary, under cold stress treatment, only *drb2* and *drb3* plants showed increased anthocyanin accumulation, while *DRB2* and *DRB3* overexpressing plants had no surplus anthocyanin production. Further analysis suggested that *DRB3* overexpression down-regulated crucial anthocyanin biosynthesis genes (such as *CHS*, *DFR,* and *ANS*) (Figure 4). Interestingly, in double mutant *DRB3ox*/*pap1-D PAP1* TF and *CHS*, *DFR* and ANS genes were also down-regulated, proposing the role of *DRB3* as a negative master regulator of stress-mediated anthocyanin biosynthesis [82].

ZIP. Basic leucine zipper (bZIP) transcription factors are another important regulator of abiotic stress responses and anthocyanin accumulation [83,84]. As was shown in Radish (*Raphanus sativus* L.), salt stress treatment down-regulated the expression of *RsbZIP059*, while *RsbZIP005* and *RsbZIP031* TFs were up-regulated. Furthermore, 39 anthocyanin biosynthesis-related genes were found to contain G-box or ACE-box elements, which could be recognised and bound by bZIP family members. These results suggested the involvement of bZIP TFs in response to abiotic stress and anthocyanin biosynthesis in radish [85].

C2H2. Zinc finger protein is one of the most widely distributed TF family proteins in eukaryotes. The C2H2-type zinc finger proteins play an important role in plant growth and development [86]. For example, the expression of apple MdZAT5 TF is induced by various stresses (such as cold, salt, high light, and ABA). Overexpression of *MdZAT5* in apple calli up-regulated the expression of anthocyanin biosynthesis-related genes (*ANR*, *CHI*, *CHS*, *DFR*, *F3H*, and *UFGT*) and caused increased anthocyanin accumulation (Figure 4). Increased anthocyanin content was also achieved in transgenic *Arabidopsis* plants, expressing *MdZAT5*. Furthermore, overexpression of *MdZAT5* in apple calli increased sensitivity to salt stress. Accordingly, the expression of *MdZAT5* in Arabidopsis reduced the expression of important salt-stress-related genes (transporter *AtNHX1* and *PP2C phosphatase AtABI1*) and improved the sensitivity to salt stress. Therefore, these data suggest that MdZAT5 acted as a positive regulator of anthocyanin biosynthesis and as a negative regulator of salt stress resistance [87].

NAC. The member of the NAC (No Apical Meristem/NAM, Arabidopsis ATAF1/2, and Cup-shaped Cotyledon2/CUC2) TF family has been identified as a negative regulator of stress-mediated anthocyanin biosynthesis [88]. Thus, *ANAC032* repressed the expression of anthocyanin biosynthesis (*DFR*, *ANS*/*LDOX*) and positive regulatory TF (*TT8*) genes in response to oxidative, high light, and sucrose stresses. Similarly, *ANAC032* overexpressing lines produced a reduced amount of anthocyanin under salt stress when compared with wild-type plants. ABA and JA plant hormones are known to induce anthocyanin biosynthesis in plants, while the accumulation of anthocyanin was reduced in the *ANAC032* overexpressing lines, suggesting that *ANAC032* also repressed the JA and ABA-induced anthocyanin biosynthesis pathways [88].

### 3.3. Manipulation with the Anthocyanin Biosynthesis Genes to Improve Salt Stress Tolerance

It is known that the expression of flavonoid pathway-related genes is strongly up-regulated under salt stress. For example, the expression of over 40 genes (including *FLS*, *F3H*, *LDOX*, *IFS,* and *CHS*) was greatly up-regulated in *Reaumuria trigyna* (Maxim.), a typical native desert halophyte [89]. Therefore, the expression of *R. trigyna* LDOX, a crucial enzyme converting leucoanthocyanidins to anthocyanidins, complemented *Arabidopsis LDOX* mutant (*testa11*), and normalised proanthocyanin and anthocyanin levels in seeds. Interestingly, RtLDOX also performed F3H dioxygenase activities, which convert naringenin to dihydrokaempferol. Finally, *RtLDOX* expressing transgenic *Arabidopsis* mutant plants demonstrated higher flavonoid content, increased antioxidant activities, and plant biomass. These results suggested that multifunctional dioxygenase properties of RtLDOX increased flavonoid biosynthesis and facilitate enhanced salt stress response [90].

Similarly, expression of the *AtDFR* gene in *Brassica napus* (L.) increased anthocyanin accumulation and reduced ROS production under salt and mannitol stresses. Additionally, shoots of the transgenic *B. napus* plants demonstrated higher chlorophyll content in the high-salt medium than wild-type plants [91]. Anthocyanin 3-O-glucosyltransferase isolated from purple maize (or *Bronze gene* (*BZ1*), a key anthocyanin biosynthesis enzyme) was shown to contribute to salt tolerance. Under salt-stress treatment, wild-type plants accumulated more anthocyanin in comparison with bronze mutants and participated in ROS scavenging, thereby enhancing tolerance to the salt stress [92].

### 3.4. Other Factors, Regulating Anthocyanin Biosynthesis in Salt Tolerance

The UV RESISTANCE LOCUS8 (UVR8) protein was identified as a crucial factor regulating UV-B-induced anthocyanin accumulation in the hypocotyls of radish (*Raphanus sativus* L.). UV-B irradiation, cold, cadmium, and salt stresses up-regulated the expression of *UVR8* and induced anthocyanin accumulation, while these effects disappeared under dark exposure, suggesting a light-dependent mechanism. Further experiments with H_2_O_2_ and NO scavengers and releasing compounds suggested that both H_2_O_2_ and NO acted as important mediators in abiotic stresses, *UVR8* expression, and anthocyanin accumulation [93].

Recently, green silver nanoparticles (GSN) derived from the leaf extract of *Hibiscus sabdarifa* (L.) were used to improve the salt stress tolerance of *H. sabdarifa* plants. Accordingly, GSN application improved seed germination, leaf and root fresh and dry weights, plant height, and relative water content under salt stress. Furthermore, GSN enhanced antioxidant enzyme activities (SOD, POD, APX, and CAT) and increased levels of osmoprotectors (soluble sugars, carbohydrates, and proline). Also, the expression of anthocyanin-biosynthesis-related genes (*ANS*, *F3H,* and *CHS*) was increased and resulted in an enhanced accumulation of flavonoids and anthocyanins [94].

Similarly, inoculation with arbuscular mycorrhizal fungi (*Glomus etunicatum*) symbionts promoted the growth of the host plant and alleviated salt stress [95]. Thus, inoculated rice plants of pigmented salt-tolerant cultivar Leum Pua demonstrated increased fructose and proline accumulation under salt stress conditions in comparison with the green salt-tolerant cultivar Pokkali. Interestingly, symbiont inoculation also increased the total anthocyanins content and accumulation of cyanidin-3-glucoside and peonidin-3-glucoside in the pericarp of Leum Pua under salt stress conditions, while these anthocyanins were absent in the Pokkali cultivar [95].

In total, many TFs of different types and other proteins have been shown to regulate salt-stress-induced anthocyanin biosynthesis and accumulation. Discussed studies provided new insights for future research on anthocyanin biosynthesis and resistance to salt stress and suggested new candidate genes for future improvements in valuable plant features. However, the exact molecular mechanism of their role in regulating anthocyanin accumulation and salt stress resistance is not clear. Further experiments with TFs—positive/negative regulators of anthocyanin biosynthesis and salt stress resistance, and TF—opposite regulators of anthocyanin biosynthesis and salt stress resistance (such as DRB3 and MdZAT5), will help to improve our understanding of the involved regulatory mechanisms and interplay with other involved factors.

## 4. Anthocyanin Biosynthesis and Tolerance to Several Stressors

Very often in nature several abiotic stresses (such as high salinity, drought, UV radiation, high light, and others) act simultaneously and negatively impact the growth and development of crops and pasture, greatly reducing the efficiency of the agricultural and animal husbandry industry and threatening the food supply of a rapidly growing global population [96]. Further in this section, we focus on the relationship between anthocyanin and tolerance to several abiotic stress factors (including drought and salt).

For example, recent research suggested anthocyanin content as a marker of drought and salt stress tolerance [97]. Accordingly, anthocyanin-rich tomato genotype LA-1996 demonstrated better tolerance to salinity and drought treatment in comparison with genotypes with lower anthocyanin content (LA-2838 and LA-2662). Also, LA-1996 plants more effectively regulated Na^+^, K^+^, and Ca^2+^ levels in leaves, accumulated more proline and demonstrated higher activity of antioxidant enzymes (SOD, POD, and CAT). Furthermore, LA-1996 plants produced less MDA and ROS under stress conditions, and their seed germination and root elongation were less affected by stresses. Finally, other stress-related TFs and proteins (SlDREB2A, SlWRKY8, SlABF4, SlAREB1, and NCED1) were up-regulated at a higher rate in comparison with other genotypes. Principal component analysis showed that leaf and stem anthocyanin content was associated with the major stress tolerance traits in LA-1996 [97].

### 4.1. Manipulation of Anthocyanin Biosynthesis Genes to Improve Both Drought and Salt Stress

Two *Arabidopsis* glycosyltransferases, UGT79B2 and UGT79B3, were shown to participate in anthocyanin metabolism by adding UDP-rhamnose to cyanidin and 3-O-glucoside-cyanidin [98]. Abiotic stresses (salt, drought, and cold) strongly up-regulated the expression of both genes (*UGT79B2* and *UGT79B3*). *ugt79b2/b3* double mutant plants showed reduced anthocyanin levels and increased sensitivity to abiotic stresses, while *UGT79B2/B3* overexpression strongly increased the anthocyanin accumulation, antioxidant activities (ascorbate and glutathione), and enhanced plant tolerance to cold, drought, and salt stresses. However, *UGT79B2/B3* overexpressing in *tt18* (*transparent testa 18*), a mutant with a blocked anthocyanin biosynthesis process, did not improve plant adaptation to stress [98].

The early discussed gene *LDOX2* from *R. trigyna* was also demonstrated to improve tolerance to several abiotic stresses through the increased accumulation of anthocyanins and flavonols. Accordingly, the transgenic *Arabidopsis* plants expressing the *RtLDOX2* gene accumulated more anthocyanins and proline, showed higher expression of antioxidants (*SOD*, *POD*, and *CAT*), and enhanced the primary root length and chlorophyll content, thus improving tolerance to salt, drought, and UV stresses [99]. Similarly, the expression of 2-oxoglutarate/Fe(II)-dependent dioxygenases (*2-ODDs*) from Antarctic moss *Pohlia nutans* ((Hedw.) Lindb.) in *Physcomitrella patens* and *A. thaliana* caused increased accumulation of anthocyanins and flavonol, improved seed germination, and exhibited longer root growth and larger gametophyte sizes. Furthermore, *2-ODD* expression in mutant plants increased antioxidant capacity and improved tolerance to salt, drought, UV, and oxidative stresses. These results suggested that 2-ODD is a crucial enzyme involved in the adaptation of *P. nutans* to harsh polar environments through anthocyanins and flavonol accumulation [100].

### 4.2. Transcription Factors Regulating Anthocyanin Biosynthesis and Resistance to Several Stressors

MYB and bHLH. Delila (Del), a bHLH family TF, and Rosea1 (Ros1), an MYB family TF, are two well-studied TFs regulating anthocyanin production. As was shown on transgenic tomato plants, DEX inducible expression of *Antirrhinum majus* (L.)-derived *Del* and *Ros1* genes enhanced expression of anthocyanin biosynthesis genes (*DFR*, *ANS*, *A3GT*, *F3H*, *F3′5′H*, and *AnthOMT*), trichome development, and root morphology, while it down-regulated auxin-related genes (*PINs* and *LAXs*) [101]. Accordingly, *AmDel* expression in tobacco (*Nicotiana tabacum* L.) leads to up-regulation of the *CHS*, *CHI*, *F3H*, *DFR*, and *ANS* genes, thus enhancing anthocyanin production and accumulation in leaf and flowers. Also, transgenic plants demonstrated higher polyphenol content, expression of antioxidants, and radical scavenging capacity, which resulted in an overall greater improvement in salt and drought tolerance than wild-type plants [102]. Similarly, the expression of *AmRos1* in tobacco up-regulated the expression of anthocyanin biosynthesis-related genes (*CHS*, *CHI*, *F3H*, *DFR,* and *ANS*) and promoted anthocyanin accumulation in leaves and flowers. Transgenic plants showed higher antioxidant activities, lower MDA content, and up-regulated expression of general abiotic stress-related genes (*CBF*, *Osmotin,* and *ABF*). Furthermore, transgenic plants demonstrated improved tolerance to cold and drought stresses, which was accompanied by a reduction in plant growth [103].

MYB3 TF from abiotic stress-resistant wild tiger lily (*Lilium lancifolium* L.) was up-regulated by various stresses (cold, drought, salt, and ABA), suggesting its important role in abiotic stress tolerance. Therefore, *A. thaliana* mutants expressing *LlMYB3* displayed the ABA hypersensitive phenotype and enhanced tolerance to salt, drought, and cold stresses. Interestingly, *LlMYB3* co-expresses and binds with the *LlCHS2* gene, thus proposing its involvement in stress-mediated regulation of anthocyanin biosynthesis [104].

Interestingly, MYB3 TF from *Fagopyrum tataricum* ((L.) Gaertn.) was characterised as a negative regulator of anthocyanins and proanthocyanidin biosynthesis. While the expression of *FtMYB3* was greatly induced by hormones (SA and JA) and abiotic stresses (cold, salt, and drought) treatment, the *FtMYB3* expressed in *Arabidopsis* down-regulated anthocyanin biosynthesis-related genes (*TT13*, *BAN*, *ANS,* and *DFR*). These results demonstrated that *FtMYB3* regulated multiple structural genes involved in flavonol biosynthesis. However, the involvement of SA and JA regulation suggested that *FtMYB3* may regulate anthocyanin and PA biosynthesis through multiple pathways [105].

DREB. The expression of *DREB3* TFs from *Ammopiptanthus mongolicus* ((Maxim. ex Kom.) S. H. Cheng), a high tolerance desert broadleaf shrub, in *Arabidopsis*, greatly improved tolerance to drought, heat, and high salinity stresses. Also, transgenic plants showed a deep purple colouration and increased anthocyanin accumulation, suggesting that *AmDREB3* TF acted as a positive regulator of both abiotic stress tolerance and anthocyanin accumulation [106].

REV. Related to ABI3/VP1 (RAV) TFs are known to play crucial roles in plant growth, development, and response to biotic and abiotic stresses [107]. Recently, the role of *RAV6* and *RAV7* TFs in response to drought and salt stress, respectively, was demonstrated in pear (*Pyrus bretschneideri* Rehd). However, only *PbRAV6* was shown to participate in the regulation of anthocyanin biosynthesis and, therefore, of pericarp colouring [108].

Myc. Some members of the Myc TF family in wheat have been characterised recently as crucial players of both abiotic stress response and anthocyanin biosynthesis. Thus, *TaMyc-B1* acted as a co-regulator of the *TaC1-A1*, a R2R3-Myb TF, part of the MBW regulatory complex which activates anthocyanin synthesis in wheat coleoptile. Drought stress treatment up-regulated expression of *TaMyc* genes (*A1*, *B1*, *A2,* and *D2*), while application of salt stress up-regulated only *TaMyc-B1* and *TaMyc-A2* genes and down-regulated *TaMyc-A1* TF [109].

Therefore, apart from the role in tolerance to drought and salt stresses, anthocyanin biosynthesis and accumulation are involved in tolerance to other types of biotic and abiotic stresses. These results suggested that anthocyanins are part of a rather unspecific general response to various stresses, which is mostly mediated through antioxidant/ROS scavenging (Figure 5). However, the role of other factors, such as involvement of a cross-talk with plant hormone signalling pathways (primarily, ABA, GA, and JA), lignin biosynthesis, and various channels (NHX, CHX, KEA, and others), require further investigation.

## 5. Conclusions and Future Prospective Studies

Plants are constantly subjected to various stress factors of both biotic and abiotic origins. Recently developed genomic and proteomic tools and methods have expanded our understanding of plant signal transmission and stress-mediated gene regulation. As a result, the role of anthocyanin biosynthesis and accumulation has been linked to stress resistance. Many studies on models and economically important plant species have proved that transgenic plants with increased anthocyanin levels have increased tolerance to various stresses. Furthermore, different aspects of anthocyanin biosynthesis and regulation have been discovered, and positive and negative regulators have been identified. However, the exact molecular mechanisms underlying enhanced stress resistance of anthocyanin-enriched plants are mostly unknown and require further exploration. In line with some studies, which found no association between anthocyanin levels and stress tolerance, this knowledge is vital for the production of stress-resistant and biofortified crops.

Although significant progress has been made in our understanding of the biological role of anthocyanins in the development of tolerance to environmental stresses, there are still many interesting questions that we need to address in future work. For example, how the anthocyanin biosynthesis regulating network is associated with known physiological stress-related responses based on the function of various transporters and channels (such as NHX, CHX, KEA, and others) (ion exclusion, stomatal closure, maintenance of Na^+^/K^+^ ration, and others). Also, TFs of many types have been shown to regulate the expression of anthocyanin biosynthesis structural and related regulatory genes. However, the characterisation of associated post-translational modifications (methylation, phosphorylation, S-nitrosylation, and so on) is still lacking. Finally, the role of plant hormone signalling networks should be better integrated into environmental stresses–anthocyanin relations. So far, the best characterised and studied plant hormone in this regard is abscisic acid, while the role of other plant hormones is mostly unknown.

Addressing these research questions will help us to gain a better understanding of the role of plant anthocyanins in the development of stress tolerance and provide necessary advantages to improve the plant breeding process, achieve economic efficiency and, subsequently, promote human health.

## Figures and Tables

**Figure 1 plants-12-02558-f001:**
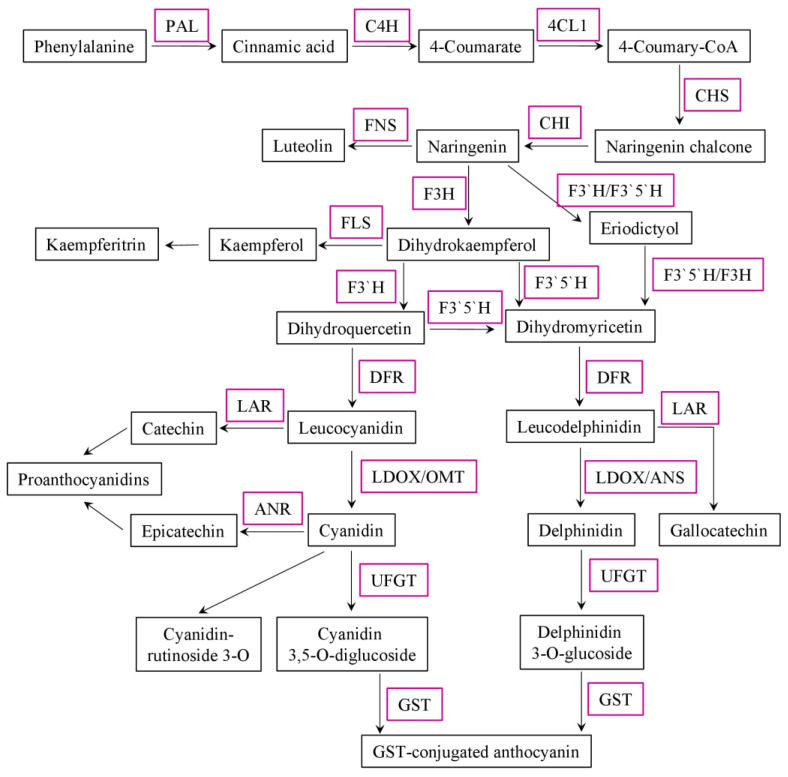
Schematic representation of the anthocyanin biosynthesis pathway. Genes in magenta boxes: PAL (phenylalanine ammonia-lyase); C4H (cinnamate-4-hydroxylase); 4CL (4-coumarate-CoA ligase); CHS (chalcone synthase); CHI (chalcone isomerase); FNS (flavone synthase); F3H (flavanone 3-hydroxylase); F3′H (flavonoid 3′-hydroxylase); F3′5′H (flavonoid 3′,5′-hydroxylase); FLS (flavonol synthase); DFR (dihydroflavonol 4-reductase); LAR (leucoanthocyanidin reductase); LDOX (leucoanthocyanidin dioxygenase); OMT (O-methyltransferase); ANS (leucoanthocyanidin dioxygenase); ANR (anthocyanidin reductase); UFGT (UDP-glucose: flavonoid 3-O-glucosyltransferase); and GST (glutathione S-transferase).

**Figure 2 plants-12-02558-f002:**
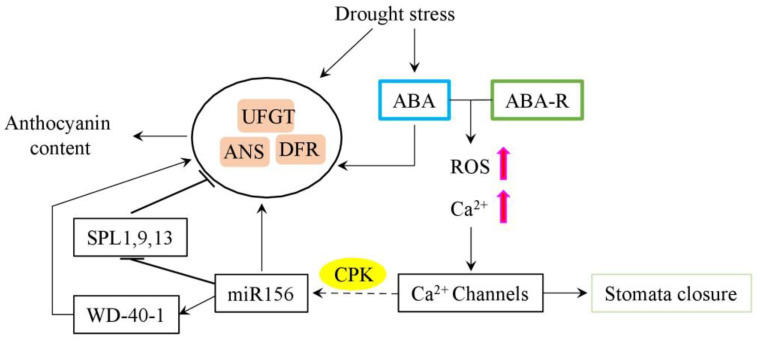
Schematic representation of miR156-ABA-based regulatory module of drought tolerance and induction of anthocyanin biosynthesis. Under drought conditions, the increased ABA level reduced stomatal aperture and promoted anthocyanin biosynthesis, thus increasing anthocyanin concentrations. Furthermore, acting via the Ca^2+^ sensor CPK, drought induced miR156, which, in turn, silenced SPLs and increased levels of WD40-1, thus enhancing anthocyanin accumulation. Arrows represent activation; blunt arrows—repression; dashed arrow—the involvement of several intermedium steps; and magenta arrows—increased levels of corresponding metabolites. Blue box—ABA (abscisic acid); green box—ABA-R (abscisic acid receptor); yellow oval—CPK (Ca^2+^-dependent protein kinase); ROS (reactive oxygen species); and SPL (squamosa promoter binding protein-like) gene.

**Figure 3 plants-12-02558-f003:**
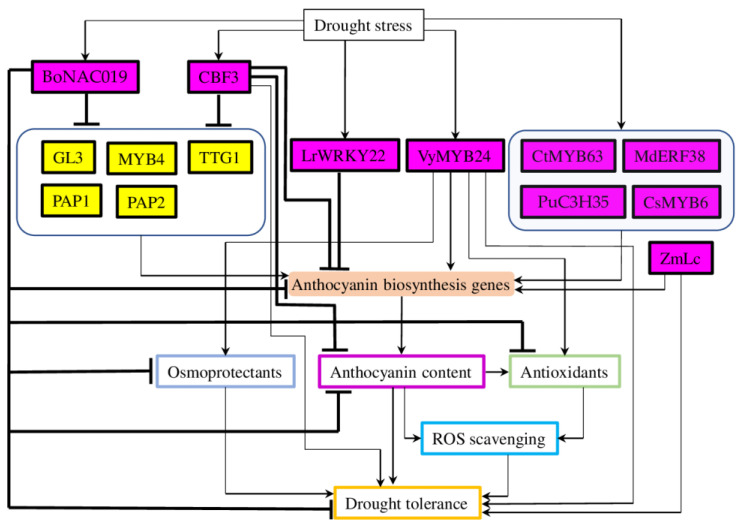
A model of TFs regulating anthocyanin biosynthesis and tolerance to drought. Drought-regulated anthocyanin production and the development of drought stress tolerance through the set of TFs—positive and negative regulators. Some TFs (such as NAC019 and CBF3) can be defined as “master regulators” because they also orchestrate the expression of other TFs regulating anthocyanin biosynthesis. NAC019 acted as a purely negative regulator of stress-induced anthocyanin biosynthesis and drought tolerance, while CBF3 negatively regulated all processes except “drought tolerance”. Magenta boxes depict TFs discussed in this section; yellow boxes—TFs known to regulate anthocyanin biosynthesis. The “Osmoprotectors” box represents proline, soluble sugars, and carbohydrates; “Antioxidants”—CAT (catalase), POD (peroxidase), SOD (superoxide dismutase), and glutathione. Arrows represent positive regulation and blunt thick lines—negative regulation. NAC (No Apical Meristem/NAM, Arabidopsis ATAF1/2, and Cup-shaped Cotyledon2/CUC2); CBF (C-repeat-binding factor); GL3 (GLABRA 3); MYB (myeloblastosis); TTG1 (TRANSPARENT TESTA GLABRA 1); PAP1 (PRODUCTION OF ANTHOCYANIN PIGMENT); ERF (ethylene response factor); C3H35 (ZINC FINGER CCCH DOMAIN-CONTAINING PROTEIN 35); and Lc (leaf colour). Taxonomic affiliation: Bo (*Brassica oleracea* L); Lr (*Lycoris radiata* (L’Hér.) Herb); Vy (*Vitis yanshanesis* J. X. Chen.); Ct (*Carthamus tinctorius* L.); Md (*Malus* × *domestic*); Pu (*Populus ussuriensis* Kom.); Sc (*Chaenomeles speciosa* (Sweet) Nak.); Zm (*Zea mays* L.); and TFs with no affiliation belong to *Arabidopsis thaliana* (L.) Heynh.

**Figure 4 plants-12-02558-f004:**
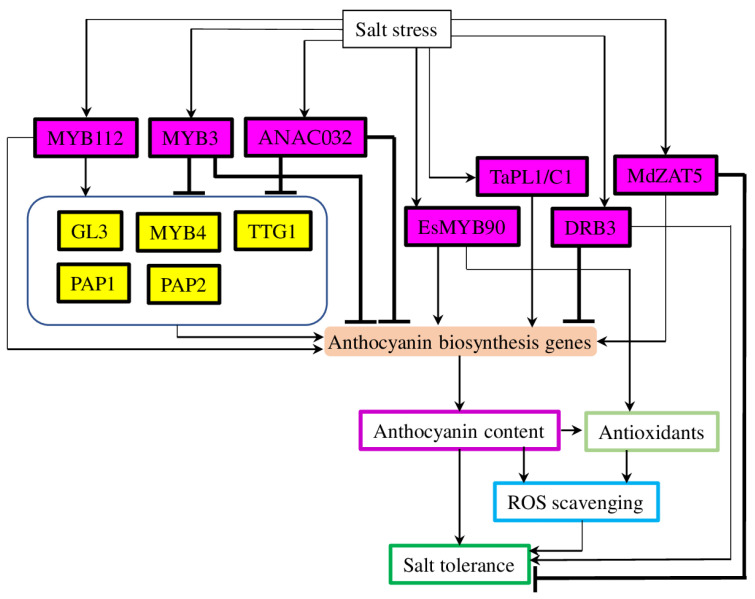
A model of TFs regulating anthocyanin biosynthesis and tolerance to salt stress. Salt stress-regulated anthocyanin biosynthesis and the development of tolerance to salt stress through the set of TFs—positive and negative regulators. DRB3 and MdZAT5 acted oppositely as a negative regulator of anthocyanin biosynthesis and a positive regulator of salt stress tolerance (for DRB3) and vice versa (for MdZAT5), respectively. MYB3 and ANAC032 acted as negative and MYB112 as positive master regulators because they regulated a set of other TFs regulating anthocyanin biosynthesis. Magenta boxes depict TFs discussed in this section; yellow boxes—TFs known to regulate anthocyanin biosynthesis. “Antioxidants”—CAT (catalase), POD (peroxidase), SOD (superoxide dismutase), and glutathione. Arrows represent positive regulation and blunt thick lines—negative regulation. GL3 (GLABRA 3); MYB (myeloblastosis); TTG1 (TRANSPARENT TESTA GLABRA 1); PAP1 (PRODUCTION OF ANTHOCYANIN PIGMENT); PL1/C1 (PURPLE PLANT1/COLORED ALEURONE1); ZAT (zinc finger of Arabidopsis thaliana); and DRB (double-stranded RNA-binding protein).

**Figure 5 plants-12-02558-f005:**
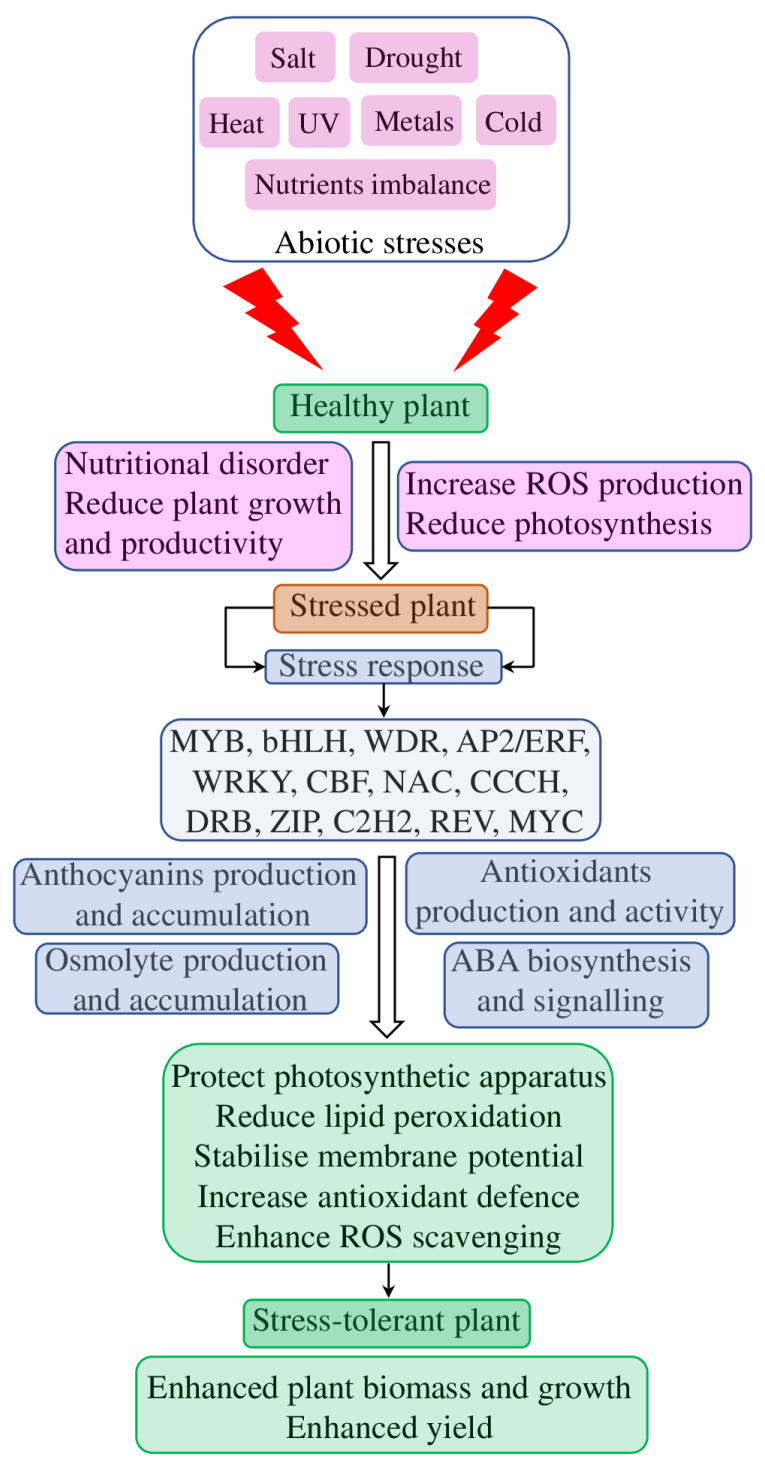
The role of anthocyanins in the development of abiotic-stress-tolerant plants. Abiotic stresses negatively affect plants and cause different physiological, biochemical, and molecular responses through activation of various transcription factors. Together with other adaptive mechanisms (such as osmolyte and antioxidant production, and increased ABA biosynthesis), increased anthocyanin biosynthesis and accumulation is aimed to reshape plant physiological, metabolic, and morphological parameters in order to make them more tolerant to various abiotic stresses and to improve their productivity.

## Data Availability

Not applicable.

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
