# Peer review of "The Role of Anthocyanins in Plant Tolerance to Drought and Salt Stresses"

_plants, 2023, doi:10.3390/plants12132558_

Round 1

Reviewer 1 Report

Dear authors, the manuscriptThe role of anthocyanins in plant tolerance to drought and salt  Stressesis quite interesting and worth investigation.Please see some comments:

1 - 2.1. Anthocyanins induction by drought stresseach paragraph should be more concise.

2 - 4.2 MYB&bHLH and MYB two parts can be merged for discussion.

5 - Figure 5 needs to be redrawn, which is not relevant to the content of the article.

Please double-check formatting and enghish grammar.

Author Response

Dear Editor and Reviewers,
We greatly appreciate your critical evaluation of our manuscript and helpful comments. Our reply to your comments would be provided point by point, where “A” stands for “Authors”, and “L” for “Lines”, where changes have been implemented. The language of the entire manuscript has been checked and corrected.

____________________________________________________________________________

Comments and Suggestions for Authors

Dear authors, the manuscript‘The role of anthocyanins in plant tolerance to drought and salt Stresses’is quite interesting and worth investigation. Please see some comments:

1 – 2.1. ‘Anthocyanins induction by drought stress’each paragraph should be more concise.

A: Thank you very much for your suggestion. We have considerably edited section 2.1. according to your remarks.

2 - 4.2 MYB&bHLH and MYB two parts can be merged for discussion.
A: Thanks. MYB&bHLH and MYB parts have been merged.

5 - Figure 5 needs to be redrawn, which is not relevant to the content of the article.
A: We would like to disagree with the reviewer. Figure 5 is central and summarises the whole content of the manuscript. Besides the drought and salt stresses (which are the major focus of the manuscript) other stressors have been also discussed in section 4. Depicted molecular, biochemical and physiological responses relevant to the role of anthocyanin in the development of stress tolerance are discussed throughout the entire manuscript.

Reviewer 2 Report

Comments to the Author

The manuscript entitled “The role of anthocyanins in plant tolerance to drought and salt stresses” explains the emerging field of anthocyanin role in the plant tolerance to the various abiotic stresses. The author has covered the all the relevant information and current research regarding the role of anthocyanin in plant tolerance to abiotic stresses like drought and salt in different crops. However, the below concerns to be addressed before acceptance of the manuscript for publication:

·         The current research didn’t provide the in depth of molecular pathways involved in physiological and metabolic responses observed in the colored drought- tolerance species, which further require research on functional characterization of genes involved in the colored drought tolerance.

·         Figure 1. Schematic representation of anthocyanin biosynthesis pathway: Explaining this pathway of biosynthesis in words with functions of all the major genes which are involved in pathway will be great helpful for the reader to understand in a better way.

·         The role of anthocyanin in crop like rice need to be added.

·         The references need to be given in accordance with journal format.

Author Response

Dear Editor and Reviewers,
We greatly appreciate your critical evaluation of our manuscript and helpful comments. Our reply to your comments would be provided point by point, where “A” stands for “Authors”, and “L” for “Lines”, where changes have been implemented. The language of the entire manuscript has been checked and corrected.

____________________________________________________________________________

Comments to the Author

The manuscript entitled “The role of anthocyanins in plant tolerance to drought and salt stresses” explains the emerging field of anthocyanin role in the plant tolerance to the various abiotic stresses. The author has covered the all the relevant information and current research regarding the role of anthocyanin in plant tolerance to abiotic stresses like drought and salt in different crops. However, the below concerns to be addressed before acceptance of the manuscript for publication:

  1. The current research didn’t provide the in depth of molecular pathways involved in physiological and metabolic responses observed in the colored drought- tolerance species, which further require research on functional characterization of genes involved in the colored drought tolerance.
    A: We would like to disagree partially with this statement.
    Indeed, further research is required. This could be used literally for any research topic.

However, several anthocyanin-related pathways, many individual genes and regulatory TFs have been discovered and characterised. In this review, we have summarised the recent papers studying various aspects of the relation between anthocyanin and tolerance to salt and drought stresses.

  1. Figure 1. Schematic representation of anthocyanin biosynthesis pathway: Explaining this pathway of biosynthesis in words with functions of all the major genes which are involved in pathway will be great helpful for the reader to understand in a better way.
    A: The figure was further explained. Because the other reviewer has asked to make the text more concise, the explanation is rather brief (please, see L32-52).

  2. The role of anthocyanin in crop like rice need to be added.
    A: Several important crops have been covered in this review (such as rice (in the last paragraph of section 3.1) wheat (5th paragraph of section 2.1, 2nd paragraph of section 2.4, 2nd paragraph of section 3.1, 1st and 2nd paragraphs of the section 3.2.1, “Myc” paragraph of section 4.2), maize (section 2.3.2 and 2nd paragraph of the section 3.3), buckwheat (3rd paragraph of the section 4.2) and others. The additional citation about rice was added (please, see 3rd paragraph of section 3.4)(L633-640).

  3. The references need to be given in accordance with journal format.

A: We would like to remind, you that Plants is a “free format submission” journal: https://www.mdpi.com/journal/plants/instructions

We have prepared our references as recommended by the publisher, the references were formatted by Zotero software and in the suggested reference style (https://www.zotero.org/styles/?q=id%3Amultidisciplinary-digital-publishing-institute).

Reviewer 3 Report

The ms plants-2477771 review the role of anthocyanins in plant tolerance to drought and salt stresses. The ms is well written, but the authors have to follow my comments to improve their ms.

The abstract should be improved and the authors should focus on the most highlighted points in their ms.

Keywords and other places: Authors do not have to add stress after drought “drought stress” because drought is known as a biotic stress without mentioning or adding the word “stress” after it. So, please revise this issue in your ms.

The authors should make all long sentences as short as they can to make it easier for the reader to follow you.

L56-57 please cite this uncited text with the following suitable ref; https://doi.org/10.3390/plants10020259

L83-91 add suitable citations

L95-104 add suitable citations

Instead of citing a specific reference at the end of the paragraph and leaving tens of lines without suitable citations, I recommend the authors to add the citation in the first sentences and then can go for more details for the same ref. For example, look for 459-470, you have not cited any suitable ref except one ref at the last sentence.

L264 Do not abbreviate the topic or titles or subtitles such as 2.3.1. MYB

L62-63 please always add citations for your text: https://doi.org/10.15835/nbha49112248

The authors have to add additional citations under each section instead of writing about only one or two references. When you write such a review, you should focus on many and many related topics and investigations that are related to your topic to make it valuable.

Also, if you presented the Latin name for a plant species, then you do not have to use the Lain name every time instead you can use the common name, the English name.

I suggest the authors to add one-two table showing the values as a reduction in growth and yield as a result of salt or drought from different investigations and the role of anthocyanins in mitigating the drought and salt stresses.

Also, I recommend the authors to add a new paragraph about the future thoughts after producing such review, what is next to be done in this area?

Good luck

Reviewer

Minor editing of English language required

Author Response

Dear Editor and Reviewers,
We greatly appreciate your critical evaluation of our manuscript and helpful comments. Our reply to your comments would be provided point by point, where “A” stands for “Authors”, and “L” for “Lines”, where changes have been implemented. The language of the entire manuscript has been checked and corrected.

____________________________________________________________________________

The ms plants-2477771 review the role of anthocyanins in plant tolerance to drought and salt stresses. The ms is well written, but the authors have to follow my comments to improve their ms.

  1. The abstract should be improved and the authors should focus on the most highlighted points in their ms.
    A: The abstract was modified, please see L10-20.

  2. Keywords and other places: Authors do not have to add stress after drought “drought stress” because drought is known as a biotic stress without mentioning or adding the word “stress” after it. So, please revise this issue in your ms.
    A: Corrected through the entire manuscript

  3. The authors should make all long sentences as short as they can to make it easier for the reader to follow you.
    A: The manuscript was edited accordingly to ensure clarity and fluency

  4. L56-57 please cite this uncited text with the following suitable ref; https://doi.org/10.3390/plants10020259
    A: the reference was added (please, see L69)

  5. L83-91 add suitable citations
    A: the reference was added (please, see L98)

  6. L95-104 add suitable citations
    A: the reference was added (please, see L109)

  7. Instead of citing a specific reference at the end of the paragraph and leaving tens of lines without suitable citations, I recommend the authors to add the citation in the first sentences and then can go for more details for the same ref. For example, look for 459-470, you have not cited any suitable ref except one ref at the last sentence.
    A: As suggested, the citations in the long paragraphs were added to the first sentences.

  8. L264 Do not abbreviate the topic or titles or subtitles such as 2.3.1. MYB
    A: The titles and subtitles were corrected throughout the entire manuscript

  9. L62-63 please always add citations for your text: https://doi.org/10.15835/nbha49112248
    A: Unfortunately, the suggested paper would not be relevant to the context – glycine-betaine was not covered in our manuscript. We would like to insist that the currently used reference (3389/fpls.2019.00080) is more suitable.

  1. The authors have to add additional citations under each section instead of writing about only one or two references. When you write such a review, you should focus on many and many related topics and investigations that are related to your topic to make it valuable.
    A: Indeed, it is important to cover “many and many related topics”. However, our primary interest was to focus on recent publications (3-5 years) related to the anthocyanin/drouth-salt topic. Unfortunately, jumping between other related topics would lead us away from our original idea and could interfere/impair the reader’s understanding.

  2. Also, if you presented the Latin name for a plant species, then you do not have to use the Lain name every time instead you can use the common name, the English name.
    A: The full Latin name was introduced only with the first description of the species, in later cases the short form was used (for example, Arabidopsis thaliana (L.) Heynh. (L274) and thaliana (L290).

  1. I suggest the authors to add one-two table showing the values as a reduction in growth and yield as a result of salt or drought from different investigations and the role of anthocyanins in mitigating the drought and salt stresses.
    A: Thank you very much for your suggestion. However, the majority of cited papers did not assess growth/yield parameters. Also, taking into account the great diversity of described species (such as model, medical or decorative plants), such a table would not provide much sense. On the other side, it is rather obvious that salt/drouth stresses will decrease/negatively affect growth/yield, while transgene will mitigate these effects. For that reason, we added the summarising Fig.5, which depicts the major biochemical, molecular and physiological roles of anthocyanin intolerance to different stresses.

  2. Also, I recommend the authors to add a new paragraph about the future thoughts after producing such review, what is next to be done in this area?
    A: We thank for the suggestion. The future prospective section was introduced (please, see L775-791).

  3. Good luck

Reviewer

A: Thank you very much for your comments and kind wishes.

Round 2

Reviewer 1 Report

The author has made modifications to the issues and provided explanations for some of them. But I think Fig5 still needs to be revised, and the current content is not suitable as a summary of the article.

Author Response

Dear Editor and Reviewers,
We greatly appreciate your critical evaluation of our manuscript and helpful comments. Our reply to your comments would be provided point by point, where “A” stands for “Authors”, and “L” for “Lines”, where changes have been implemented.

____________________________________________________________________________

The author has made modifications to the issues and provided explanations for some of them. But I think Fig5 still needs to be revised, and the current content is not suitable as a summary of the article.

A: We sincerely appreciate the suggestion provided by the reviewer. In our efforts to enhance the clarity of the summary, we have made further modifications to Figure 5. We would like to hope that the latest version of Figure 5 will effectively address the expected changes from the reviewer.

Reviewer 3 Report

Ms has been improved 

Minor revision for English is needed 

Author Response

Dear Editor and Reviewers,
We greatly appreciate your critical evaluation of our manuscript and helpful comments. Our reply to your comments would be provided point by point, where “A” stands for “Authors”, and “L” for “Lines”, where changes have been implemented.

____________________________________________________________________________

Minor revision for English is needed 

A: Thanks. The language of the entire manuscript was further corrected.
